# PM_2.5_-Associated Hospitalization Risk of Cardiovascular Diseases in Wuhan: Cases Alleviated by Residential Greenness

**DOI:** 10.3390/ijerph20010746

**Published:** 2022-12-31

**Authors:** Haomin Yang, Jianpeng Liao, Jing Wang, Can Yang, Kuizhuang Jiao, Xiaodie Wang, Zenghui Huang, Xuxi Ma, Xingyuan Liu, Jingling Liao, Lu Ma

**Affiliations:** 1Department of Biostatistics, School of Public Health, Wuhan University, Wuhan 430071, China; 2Wuhan Information Center of Health and Family Planning, Wuhan 430021, China; 3Department of Nutrition and Food Hygiene, School of Public Health, Medical College, Wuhan University of Science and Technology, Wuhan 430081, China

**Keywords:** cardiovascular disease, fine particulate matter, residential greenness, case-crossover study

## Abstract

PM_2.5_, a type of particulate matter with an aerodynamic diameter of less than 2.5 μm, is associated with the occurrence of cardiovascular diseases (CVDs), while greenness seems to be associated with better cardiovascular health. We identified 499,336 CVD cases in Wuhan’s 74 municipal hospitals between 2017 and 2019. A high-resolution PM_2.5_ model and a normalized difference vegetation index (NDVI) map were established to estimate individual exposures. The time-stratified case-crossover design and conditional logistic regression models were applied to explore the associations between PM_2.5_ and CVDs under different levels of environmental factors. Greenness could alleviate PM_2.5_-induced hospitalization risks of cardiovascular diseases. Compared with patients in the low-greenness group (ER = 0.99%; 95% CI: 0.71%, 1.28%), patients in the high-greenness group (ER = 0.45%; 95% CI: 0.13%, 0.77%) showed a lower increase in total CVD hospitalizations. After dividing the greenness into quartiles and adding long-term PM_2.5_ exposure as a control factor, no significant PM_2.5_-associated hospitalization risks of CVD were identified in the greenest areas (quartile 4), whether the long-term PM_2.5_ exposure level was high or low. Intriguingly, in the least green areas (quartile 1), the PM_2.5_-induced excess risk of CVD hospitalization was 0.58% (95% CI: 0.04%, 1.11%) in the long-term high-level PM_2.5_ exposure group, and increased to 1.61% (95% CI: 0.95%, 2.27%) in the long-term low-level PM_2.5_ exposure group. In the subgroup analysis, males and participants aged 55–64 years showed more significant increases in the PM_2.5_-induced risk of contracting CVDs with a reduction in greenness and fine particle exposure conditions. High residential greenness can greatly alleviate the PM_2.5_-induced risk of cardiovascular admission. Living in the areas with long-term low-level PM_2.5_ may make people more sensitive to short-term increases in PM_2.5_, leading to CVD hospitalization.

## 1. Introduction

Cardiovascular diseases (CVDs), as the major chronic non-communicable diseases, have become the leading cause of death worldwide [1]. In China, the number of residents affected by CVDs reached 290 million [2]. The case number of the most serious CVD events (i.e., acute myocardial infarction and stroke) is expected to be increased over 21 million in the next 20 years due to an aging population, urbanization, and increased CVD risk factors [3].

Air pollution is one of the greatest environmental and public health threats of our time [4]. Particulate matter (PM) especially arouses public health concerns because of its toxicity and the widespread human exposure to this pollutant [5]. Previous epidemiological studies conducted on fine particulates and CVDs have shown an apparent association between PM_2.5_ exposure and the morbidity and mortality of CVDs. Two cohort studies using Cox regression included 512,689 adults (from 2004 to 2017) and 116,972 adults (from 2000 to 2015), respectively, and reported excess risks of 4.0% (95% CI: 2.0%, 7.0%) [6] and 25.1% (95% CI: 22.0%, 28.3%) [7] in CVD admissions per 10 μg/m^3^ increase in PM_2.5_. However, these studies mostly adopted the multi-angle implementation of atmospheric correction AOD to assess PM_2.5_ concentrations, ignoring the influence of land use type, transportation road, and other surface factors on individual exposure. In addition, these studies reported varying effect estimates. Researchers have often simply attributed these inconsistencies to different study designs, population characteristics, and countries, while ignoring the complex environmental exposures [8] (including greenness, concentration of pollutants, temperature, and noises) to the study participants.

Some studies have investigated the health benefits of green space. A meta-analysis that included 103 observational and 40 interventional studies demonstrated that increased greenspace exposure was associated with decreased risk of cardiovascular mortality (OR = 0.84; 95% CI = 0.76, 0.93) [9]. However, only a few studies have investigated the modification effects of greenness on the health hazards of air pollution, and the results have been mixed. Whether or to what extent the benefits of green space come about by reducing the harmful effects of air pollution remains unknown. A time series study using the daily number of hospital admission for analysis found that the association between air pollution (PM_10_, PM_2.5_) and hospitalization of cardiovascular disease was lower in areas with more green space [10]. On the contrary, a study constructed an open cohort consisting of 63 million Medicare beneficiaries, and it reported a significantly higher effect of PM_2.5_ on CVDs in greener areas when Cox-equivalent Poisson models were used for analysis [8]. Most previous studies used a regional average NDVI as the green space index, which ignored the differences in individual green space exposure, and the use of specific populations as research objects may increase the selection bias. In addition, both long-term and short-term exposures to fine particulates have been proven to be connected with increases in the morbidity and mortality of CVDs [11,12]. However, most studies have only examined the effects of a single environmental factor on CVDs which may overestimate the risk values, and further studies are needed to explore the potential modification effects between them.

Therefore, we selected residential greenness and long-term PM_2.5_ concentration as exposures, and conducted a study on PM_2.5_-induced CVD hospitalization risks under different conditions of dual environmental factors based on the admission data of all hospitals in Wuhan. Furthermore, LUR models and NDVI maps were used to better capture individual exposure. This study aimed to assist in further interpreting the complex impacts of environmental exposure and improving our understanding of the acute effects of fine particulates on CVD hospitalization.

## 2. Method

### 2.1. Study Area and Population

Wuhan is a megacity in central China, with a land area of 8569.15 km^2^ and a population of more than 10 million. We obtained hospitalization records from the Wuhan Health and Family Planning Information Center from 2017 to 2019. A total of 74 municipal hospitals were included in this study, and it covered nearly all hospitals in Wuhan that can receive inpatients. Sex, age, dates of admission and discharge, patient’s home addresses, and primary diagnoses were extracted from each record. Cardiovascular hospital admissions were identified based on the primary diagnosis according to the International Classification of Diseases, 10th Revision (ICD-10). The following admissions were included: CVD (I00−I99), hypertension (I10−I15), coronary heart disease (I20−I25), and stroke (I60−I69). A total of 6,303,083 hospital admission records were collected from 2017 to 2019. After excluding cases with home addresses outside Wuhan and non-first-time CVD hospitalizations, we identified 499,336 CVD admissions in total. The specific inclusion and exclusion criteria are outlined in Appendix A.

### 2.2. Air Pollution Assessment

Air pollution data were collected from the Wuhan Environmental Protection Bureau. The daily 24 h average concentration data for PM_2.5_ (μg/m^3^) from 2016 to 2019 were collected from 21 air-quality-monitoring stations. The concentrations of missing days were replaced by the average concentrations of each site for the month. In addition, we collected daily meteorological data from 2016 to 2019 from the China Meteorological Data Network.

Land use regression (LUR) models were used to estimate the PM_2.5_ concentrations in each patient’s residential area. In brief, we identified many potential predictor variables based on our previous study [13], such as traffic indicators, population densities, surrounding land use, industrial pollution sources, and altitude data.

Firstly, we set up a multi-ring buffer at 21 pollutant monitoring stations to extract these potential predictors through geographic information system (ArcGIS). Multiple linear regression models were adopted to fit these predictor variables with the average PM_2.5_ concentrations at the monitoring stations from 2016 to 2019. Considering that these predictor variables change over time, and that too long a time span can reduce the accuracy of the model, we built two models to simulate PM_2.5_ concentration in the 2016–2017 years (R^2^ = 0.78) and 2018–2019 years (R^2^ = 0.86), respectively. Details are shown in Appendix A.

Secondly, a PM_2.5_ concentration map of 1 km spatial resolution (Figure 1b) was established by extracting the corresponding predictor variables in each grid point, calculating the predicted PM_2.5_ concentrations, and using the Kriging interpolation method.

Finally, through the established PM_2.5_ map and individual home address, we extracted the annual PM_2.5_ concentration of each patient. Following the method described in previous studies [14], the average PM_2.5_ concentration in the year prior to the day of admission (long-term PM_2.5_ exposure) and PM_2.5_ on the day of admission (short-term PM_2.5_ exposure) were extrapolated from annual PM_2.5_ concentrations in the LUR model, adjusted for the ratio of yearly specific or daily specific PM_2.5_ to the estimated annual average PM_2.5_ at the nearest monitor.

### 2.3. Residential Greenness Assessment

The residential surrounding greenness was represented as the normalized difference vegetation index (NDVI). The data were derived from moderate-resolution imaging spectroradiometer (MODIS) images collected by NASA’s Terra satellite. The NDVI ranges from −1 to 1, and higher positive values represent higher greenness.

The MODIS captured the density of green vegetation at a spatial resolution of 250 m every 16 days since 2000 (product number: MOD13Q1). First, we downloaded all 16-day NDVI-gridded observations from 2017 to 2019, 70 in total, and combined them in each natural year. Second, the negative NDVI pixel values were transformed to zero; thus, the negative values would not offset the positive values in calculating the area average [15]. Third, the surrounding NDVI values within the different radial buffers (250 m and 500 m) centered on each participant’s home address were calculated using ArcGIS 10.7, i.e., the average density of green vegetation within a circular buffer around a participant’s residential address.

### 2.4. Study Design

A time-stratified case-crossover study design was used to evaluate the acute effects of PM_2.5_ exposure on cardiovascular hospitalization in Wuhan. The case day was defined as the day of hospital admission, and the control days shared the same year, month, and day of the week with the case day [14]. Each case was used as his or her own control by assessing the concentration of pollutants on the control day; thus, the potential confounding effects of factors that remained constant on the case and control days were largely eliminated. Finally, 1,696,434 control days were selected from the 499,336 cardiovascular admission cases.

### 2.5. Analytic Model

A conditional logistic regression (CLR) model was used to obtain estimates of the percent of excess risks (ER%) and 95% confidence intervals (CIs) for the acute effects of PM_2.5_ on CVD hospitalization (ER% = [OR−1] × 100%). The variables that fluctuate daily, such as temperature (°C) and relative humidity (%), were controlled as covariates. The natural cubic spline (NCS) function with three degrees of freedom (df) was used for temperature and humidity [16].
(1)$$\ln\left(h\left(t,x\right)\right)=\ln\left(h_{0}\left(t\right)\right)+\beta *PM_{2.5}+ns\left(\;\mathrm{temp},df=3\right)+ns\left(rh,df=3\right)$$
(2)$$ER\%=\left(e^{\beta }-1\right)*100\%$$

$\ln\left(h\left(t,x\right)\right)$ represents the hazard function, $\ln\left(h_{0}\left(t\right)\right)$ represents the baseline hazard, ns represents the natural cubic spline function, df represents the degree of freedom, and β represents the relative coefficient.

Considering that a single-day lag model might underestimate the association [17], both the concentrations of PM_2.5_ at single-lag (lag0 to lag3) and multi-day moving averages (lag0–1 to lag0–3) before admissions were analyzed independently in the separate model, which has the same parameter settings as the main model. The multi-day lag concentration was calculated according to the previous study. For example, lag0–2 was the moving average concentration on the present day and the previous 2 days (an average of lag0, lag1, and lag2).

We hypothesized that people who lived with different levels of residential exposures (i.e., the levels of long-term PM_2.5_ exposure and residential greenness) would have different sensitivities to short-term increases in PM_2.5_, or that the underlying risks of PM_2.5_ would be different. To explore this phenomenon, we first classified participants into four groups based on a single environmental factor (high/low greenness and high/low long-term PM_2.5_) to explore the potential modification effects. After then, we classified participants by dual environmental factors (both greenness and long-term PM_2.5_) to explore the robustness of the modification effects of greenness under different levels of long-term PM_2.5_. We calculated excess risks (ER%) and 95% CIs in each group separately. Furthermore, to explore individual-level effect modifiers, we conducted a stratified analysis by classifying all CVD inpatients based on age (<45, 45–54, 55–64, and >64 years) and sex (male and female) in two groups, as well as G_H_E_L_ (high greenness and low long-term PM_2.5_) and G_L_E_H_ (low greenness and high long-term PM_2.5_).

All the analyses were conducted using R, version 4.1.0. The “Survival” package was used for the CLR analysis. Population characteristics were described using percentages, means, and standard deviations. All results of the model estimates are reported as percent of excess risk (ER%) and 95% CIs associated with each 10 μg/m^3^ increase in short-term PM_2.5_ concentrations.

### 2.6. Sensitivity Analysis

Several sensitivity analyses were performed to check the robustness of main results. (1) The average PM_2.5_ concentration in the six months before admission (such as long-term PM_2.5_ exposure) and NDVI values within 500 m were used to repeat the primary analysis. (2) A symmetric bidirectional case-crossover (CCO) design (days: ±7, 14) was used to check whether different control day selections affect the results [18]. (3) The degree of freedom (df) in the natural cubic spline (NCS) function was changed for the meteorological variables (3–6 df).

## 3. Results

### 3.1. Study Population

The basic characteristics of all the cardiovascular admissions were sub-grouped using the levels of residential greenness or the long-term PM_2.5_ concentration, as detailed in Table 1. A total of 499,336 patients who lived in Wuhan and were hospitalized for CVDs between 1 January 2017 and 31 December 2019 were enrolled. The mean age of all admissions was 65.13 years (SD = 14.84). Patients with coronary heart disease (CHD) and stroke each accounted for approximately one-third of all cardiovascular inpatients. In the total and subgroups, the proportion of males was higher than that of females. Participants over 74 years of age accounted for the largest proportion. There were no statistical differences between each subgroup among the above population characteristics, which facilitated comparability between groups.

### 3.2. Residential Greenness and PM_2.5_ Assessment

The map showing the NDVIs and PM_2.5_ concentrations simulated by the LUR model is shown in Figure 1. The levels of greenness surrounding the residential area varied significantly. For example, the NDVI within 250 m of the cardiovascular admissions’ residential area ranged from 0.001 to 0.986 (median: 0.43; IQR: 0.14) (Figure 1a). The values and distributions of the NDVI in Wuhan changed slightly in 2017, 2018, and 2019, and the average annual NDVI values were 0.66, 0.67, and 0.64, respectively.

The PM_2.5_ data used for the LUR modeling in this study included the years 2016, 2017, 2018, and 2019. Therefore, for patients admitted in 2017, we were able to calculate their long-term PM_2.5_ concentrations in the year prior to admission. Figure 1b shows the map of PM_2.5_ concentration modeled by LUR. The average concentrations of estimated PM_2.5_ for short-term and long-term exposures were 44.8 μg/m^3^ (SD = 27.8 μg/m^3^) and 46.9 μg/m^3^ (SD = 4.7 μg/m^3^), respectively (Appendix A).

### 3.3. Primary Model Results

A lag analysis was conducted between PM_2.5_ and cardiovascular admissions (Table 2). The highest risk estimates were found at lag2 and lag0–2 for the single-day lags and multi-day lags, respectively. Moreover, the cumulative lag model typically had higher estimates than that of the single-day exposure. Thus, in the following analyses, we focused on lag0–2 as the exposure period to evaluate the short-term effects of PM_2.5_.

As shown in Table 2, the short-term effects of PM_2.5_ exposure were significant in the total CVD admissions at all lag models except lag3. The ER% per 10 μg/m^3^ increase in lag0–2 of PM_2.5_ for the total CVD admissions was 0.78% (95% CI: 0.58%, 0.99%). For the specific subtypes, the PM_2.5_-induced hospitalization risk of CHD was particularly pronounced, and each 10 μg/m^3^ increase in PM_2.5_ at lag0–2 was associated with a 1.01% (95% CI: 0.62%, 1.39%) increase in hospital admissions for CHD. All the effect estimates remained robust when we used a symmetric case-crossover design (days: ±7, 14) or changed the degree of freedom (df) in the natural cubic spline (NCS) function for the meteorological variables from 3 to 6 (Appendix A).

### 3.4. Modification Effects of Greenness

Table 3 shows the modification effects of residential exposure factors (greenness and long-term PM_2.5_) on PM_2.5_-associated CVD hospitalization risks. Intriguingly, we found that greenness alleviates PM_2.5_-induced CVD risks; in addition, lower and insignificant increases in the hospital admissions of total CVD events and subtypes of CVD were found in the high-greenness groups compared to the low-greenness groups (Table 3). For example, every 10 μg/m^3^ increase in lag0–2 of PM_2.5_ was associated with a 0.99% increase in total CVD hospital admissions (95% CI: 0.71%, 1.28%) in the low-greenness area and a 0.45% increase (95% CI: 0.13%, 0.77%) in the high-greenness area. After using the NDVI within 500 m as residential greenness, the alleviating effects of greenness remained robust in total CVD events and subtypes of CVDs (Appendix A). On the other hand, no significant change in estimated effects was identified in the total CVD admissions between the low and high long-term PM_2.5_ exposure groups.

We then classified the participants by using both the NDVI quartiles and the median of the long-term PM_2.5_ (Table 4). In general, the modification effects of greenness on total CVD and CHD admissions were still robust regardless of high or low levels of long-term PM_2.5_ exposure. The effect estimates for the increases in hospitalization became lower and insignificant in greener areas (quartile 3 and quartile4) compared to less green areas (quartile1 and quartile2). With the increase in greenness, the excess risk of CVD hospital admissions declined from 1.61% to 0.39% (Q1 to Q4) in low-level long-term PM_2.5_ areas and 1.19% to 0.01% (Q2 to Q4) in high-level long-term PM_2.5_ areas. In the greenest areas (quartile 4), no significant increase in PM_2.5_-induced hospitalization risks for total CVD events or subtypes of CVD was identified. Moreover, a significant modification effect of long-term PM_2.5_ exposure was found in the least green areas (quartile 1) (Table 4). Every 10 μg/m^3^ increase in lag0–2 of PM_2.5_ was associated with 1.98% (95% CI: 0.39%, 3.60%) and 1.32% (95% CI: 0.20%, 2.61%) increases in hospital admissions for hypertension and stroke cases in areas with long-term low-level PM_2.5_ exposure and the least amount of greenness. However, no significant increases were found in the corresponding high-level long-term PM_2.5_ exposure areas. We reached the same conclusion as above when the NDVI within 500 m and six-month long-term concentrations were used to define dual environmental factors (Appendix A).

### 3.5. Stratification Analysis

The subgroup analyses stratified by age and gender were conducted in G_H_E_L_ and G_L_E_H_. The PM-induced risks of CVD were found to be higher in G_L_E_H_ (areas with low greenness and long-term high-level PM_2.5_) than in G_H_E_L_ (areas with high greenness and long-term low-level PM_2.5_) in nearly all gender and age groups (Figure 2 and Appendix A). In the age-specific analysis, risk estimates for the total and cause-specific CVD admissions increased significantly from the G_H_E_L_ to the G_L_E_H_ in the group aged 55–64 years, and the increase for the total CVD admissions was 1.29% (95% CI: 0.56%, 2.02%) in the G_L_E_H_ and further declined to −0.84% (95% CI: −1.72%, 0.05%) in the G_H_E_L_. In the gender-specific analysis, the difference in the risk estimates between the G_H_E_L_ and the G_L_E_H_ was more significant in males than in females. The increase for total CVD admissions in the male group was 0.97% (95% CI: −0.48%, 1.47%) in the G_L_E_H_ and declined to 0.31% (95% CI: −0.29%, 0.91%) in the G_H_E_L_. Moreover, the same sensitivity group of age (55–64 years) was identified in the stratified analysis conducted in high-greenness and low-greenness groups (Appendix A).

## 4. Discussion

To our knowledge, this is one of the few studies to evaluate the acute effects of PM_2.5_ on CVD hospital admissions based on an entire population of a megalopolis. The results of our study showed a significant PM_2.5_-induced risk of total CVD events and specific subtypes (i.e., hypertension, CHD, and stroke) of CVD. Regarding the residential exposures, greenness has a strong alleviating effect on CVD admission risks attributed to PM_2.5_ regardless of long-term high or low levels of PM_2.5_ exposure.

Our results on the adverse effects of PM_2.5_ exposure on CVD admission are consistent with several studies which focus on other areas of China [6] and developed countries [19,20]. Our result of the lag analysis showed that the effect estimates were higher when using the moving average lag model than that of single-day exposure, with the highest effects found in lag0–2. Similar to previous studies conducted in England [20] and Denmark [21], the highest harmful effects of particulate matter on CVD admissions were found at lag0–5 and lag0–4, respectively. A toxicological study reported that Interleukin (IL)-6 and tumor necrosis factor alpha showed a significant increase with a lag of two days in association with an increase in PM_2.5_. This result may suggest that consecutive days of air pollution were connected with a greater risk of cardiovascular admissions. Of note, the short-term effects of PM_2.5_ on CHD (1.01%, 95% CI, 0.62–1.39) seemed to be the greatest in our study. Mechanistic studies have demonstrated that PM can accelerate the progression of atherosclerosis; promote the formation of thrombosis; and increase the susceptibility to CHD by increasing oxidative stress, endothelial dysfunction, and blood viscosity [22,23].

Our findings on the modification effects of greenness are novel. The PM_2.5_-induced increase in CVD hospitalizations became lower and insignificant in greener areas compared to less green areas. Moreover, such alleviating effects were found in areas with both low and high long-term PM_2.5_ concentrations. This may indicate that the modification effects of green space are not, or at least not only, generated by reducing air pollution concentrations. Similar modifying effects of greenness on the association of PM_2.5_ and CVD have been previously reported. A study conducted in the USA that included 364 counties found that the association between air pollution and cardiovascular hospitalization was less significant in areas with higher residential greenness levels, with a 0.18% decrease in the effect estimates (95% CI: −0.39, 0.73) when the NDVI increased with an interquartile range [10]. Yitshak et al. proved that the effects of PM_2.5_ on CVD mortalities were attenuated by higher residential green space levels in areas with a lower socioeconomic status [24]. In addition, a study conducted by the Chinese Elderly Health Service Cohort from 1998 to 2011 found that the PM-induced mortality risk of pneumonia showed a decreasing trend (*p* for trend = 0.034) as the green quartile increased from quartile 1 (lowest) to quartile 4 (highest) [25]. To our knowledge, possible mechanisms by which green space exerts its health benefits are as below. Urban trees and shrubs could mitigate personal, home-indoor, and home-outdoor PM levels by serving as efficient biological filters and accelerating the deposition of pollutants [26,27,28]. Residential greenness may provide a buffer against the negative health impact of stressful life events, as well as promote outdoor physical activity and social contact [29,30]. Studies have demonstrated that urban parks had a cooling effect on the surrounding environment and reduce heat exposure [31]. This might be important because warmer urban climates have been proven to increase the risks of air-pollution-related diseases [32].

Regarding the modifying effects of the long-term PM_2.5_ exposure, the acute effects of PM_2.5_ on hypertension and stroke were stronger in areas with long-term low-level PM_2.5_ exposures when the NDVI was limited to quartile 1. The results from other studies are also consistent those reported in our study. In a study of 652 cities, stronger associations were observed between particulate matter and all-cause mortality in areas with lower annual mean concentrations of PM and higher annual mean temperature [5]. Similarly, Chen R et al. reported that the cities in China with low levels of PM_10_ (from 52 to 98 μg/m^3^) exhibited a risk estimate value that is two to three times higher than those with middle (from 101 to 121 μg/m^3^) or high (130 to 144 μg/m^3^) levels of PM_10_ [12]. This can be explained by the fact that the adaptive response to PM in populations living in areas with high levels of PM long-term exposures may lead to smaller estimate changes per unit [5]. This finding may partly explain the higher rates of CVD admission in developed countries.

The stratified analysis of age and gender showed stronger modifying effects of residential environments in participants aged between 55 and 64 years and in males. Vienneau et al. found a protective effect of residential greenness on the natural-cause mortality that was stronger in younger individuals and males in Switzerland [33]. Their findings did not contradict our results since we focused on CVDs, which is a major health concern among the aging population. Additionally, we focused on hospitalization rates instead of mortality rates. Heo et al. conducted a study on the hospitalization of older people in the USA and found that the 65–74 years age group might have the biggest decrease in PM_10_-related hospitalization risk [10], and this result was similar to our findings. In addition, males generally engage in more weekly physical activity outside [34], which may be why they benefit more from a better residential environment [35].

Considering that previous studies were primarily conducted in developed countries with exposure to low-level PM, the use of Wuhan as a research area provides possible ways of understanding the acute effects of PM and the modifying role of residential environmental factors in a wide range of PM concentrations. Several approaches were used to reduce the study bias and improve accuracy and precision. First, we obtained the hospitalization admission data of 74 hospitals in Wuhan to evaluate PM_2.5_ admission relationships. The possibility of selection bias was minimized, and the results could be generalized to the entire city and all age groups. Second, a case-crossover design with a large sample size was used that was controlled for long-term trends and known or unknown time-invariant covariates (e.g., socioeconomic status and genetics). Third, the application of the LUR model improved the accuracy of the spatial variation of the individual PM_2.5_ exposure.

This study had several limitations as well. First, we were unable to directly measure personal air pollution and greenness exposure levels. Alternatively, we estimated the individual air pollution exposure concentrations and residential greenness by using the home address of each participant. This may lead to exposure misclassification. Second, we measured residential greenness via the NDVI which represents the quantity but not the quality of greenness space. Third, we used inpatient data rather than outpatient data, and this might exclude some patients with mild symptoms, thus underestimating the risk of PM_2.5_ exposure. Fourth, only two residential environmental exposure factors (residential greenness and PM_2.5_) were evaluated in this study, and the confounding effects caused by other environmental factors such as noise and air pollution emission sources could not be controlled. Further studies are needed to identify and demonstrate the alleviating effects of greenness on air pollution hazards.

## 5. Conclusions

We found that short-term exposure to PM_2.5_ was significantly associated with an increased risk of CVD admission, hypertension, stroke, and CHD hospitalizations. Residential greenness could mitigate the negative effects of PM_2.5_ on CVDs, and the modification effects were robust regardless of long-term high or low levels of PM_2.5_ exposure. In addition, living in areas with long-term low-level PM_2.5_ concentrations and low greenness may make people more sensitive to short-term increases in PM_2.5_, leading to CVD admissions. Our findings add more evidence that helps us understand the acute adverse effects of PM_2.5_ and the modifying effects of residential environmental exposure. The results of this study can assist public agencies in developing strategies for air pollution control, disease prevention, and green construction.

## Figures and Tables

**Figure 1 ijerph-20-00746-f001:**
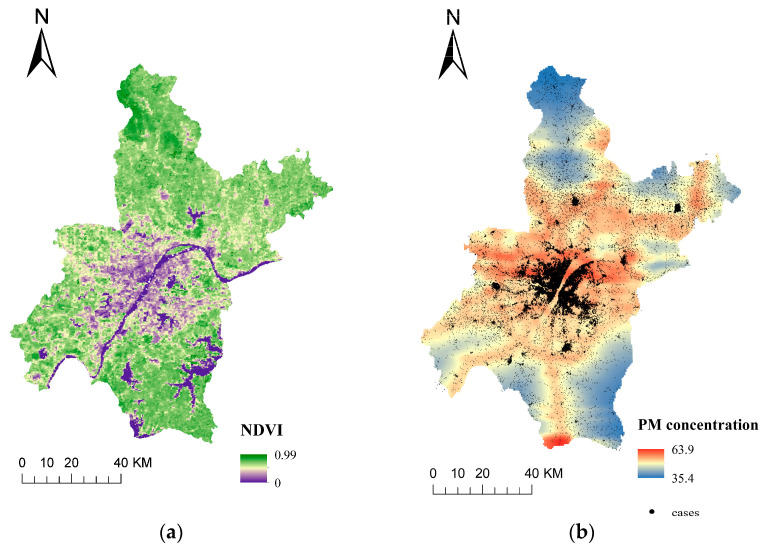
The normalized difference vegetation index (NDVI) of Wuhan measured from a 250 m resolution in 2017 (**a**). The spatial distribution of PM_2.5_ estimations across Wuhan City from 1 January 2018 to 31 December 2019 and the geocoded residential locations of the cardiovascular inpatients (**b**).

**Figure 2 ijerph-20-00746-f002:**
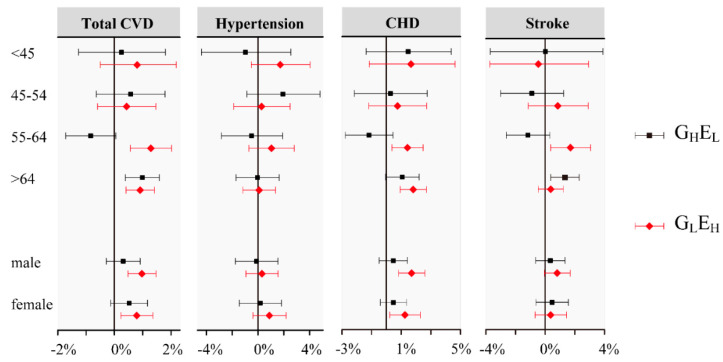
ER% (95% CI) of hospitalization risks for cardiovascular diseases per 10 μg/m^3^ increase in the lag0–2 of PM_2.5_ stratified by age and gender in G_H_E_L_ and G_L_E_H_. Abbreviation: G_H_E_L_, high-greenness and long-term low-level PM_2.5_ exposure; G_L_E_H_, low-greenness and long-term high-level PM_2.5_ exposure.

**Table 1 ijerph-20-00746-t001:** Basic characteristics of cardiovascular disease admissions in Wuhan (1 January 2017 to 31 December 2019).

Characteristics	Cardiovascular Disease (*n* = 499,336)				
	All	^a^ Low Greenness	High Greenness	^b^ High Long-Term PM_2.5_	Low Long-Term PM_2.5_
	(*n* = 499,336)	(*n* = 249,438)	(*n* = 249,898)	(*n* = 249,337)	(*n* = 249,999)
Age [mean ± SD (years)]	65.13 ± 14.84	65.48 ± 14.84	64.79 ± 14.84	65.03 ± 14.74	65.24 ± 14.95
Age group [*n* (%)]					
<45 years	41,027 (8.2)	19,573 (7.9)	21,454 (8.6)	20,460 (8.2)	20,567 (8.2)
45–54 years	65,386 (13.1)	31,981 (12.8)	33,405 (13.4)	32,181 (12.9)	33,205 (13.3)
55–64 years	124,713 (24.9)	63,972 (25.7)	60,741 (24.3)	61,983 (24.9)	62,730 (25.1)
65–74 years	128,791 (25.8)	61,506 (24.7)	67,285 (26.9)	67,633 (27.1)	61,158 (24.5)
>74 years	139,419 (27.9)	72,406 (29.0)	67,013 (26.8)	67,080 (26.9)	72,339 (28.9)
Sex [*n* (%)]					
Male	273,082 (54.6)	136,711 (54.8)	136,371 (54.6)	135,572 (54.4)	137,510 (55.1)
Female	226,254 (46.4)	112,727 (46.2)	113,527 (45.4)	113,765 (45.6)	112,489 (44.9)
^c^ Sub-diagnoses [*n* (%)]					
Hypertension	76,965 (15.4)	39,688 (15.9)	37,277 (14.9)	36,234 (14.5)	40,731 (16.3)
CHD	144,536 (28.9)	75,069 (30.1)	69,467 (27.8)	69,780 (28.0)	74,756 (29.9)
Stroke	164,390 (32.9)	79,737 (32.0)	84,653 (33.9)	86,573 (34.7)	77,817 (31.1)

^a^ Low- and high-greenness levels were divided by the median of all participants’ NDVIs within 250 m. ^b^ Low and high long-term PM2.5 values were divided by the median of all participants’ long-term PM2.5 concentrations (1 year). ^c^ Hypertension (I10–I15). CHD, coronary heart disease (I20–I25). Stroke (I60–I69).

**Table 2 ijerph-20-00746-t002:** ER% and 95% CI of hospitalization risks for cardiovascular diseases associated with every 10 μg/m3 increase in short-term PM2.5 using different lag days.

Lag	Total CVD	Hypertension	CHD	Stroke
lag0–3	**0.64**(**0.42,0.86**)	0.29(−0.26,0.84)	**0.9**(**0.48,1.31**)	**0.58**(**0.18,0.97**)
lag0–2	**0.78** (**0.58, 0.99**)	**0.53** (**0.03, 1.04**)	**1.01** (**0.62, 1.39**)	**0.69** (**0.32, 1.06**)
lag0–1	**0.57** (**0.38, 0.75**)	0.26 (−0.19, 0.72)	**0.75** (**0.41, 1.10**)	**0.58** (**0.25, 0.91**)
lag3	−0.07(−0.23,0.08)	−0.17(−0.56,0.23)	0.05(−0.25,0.34)	−0.07(−0.35,0.22)
lag2	**0.48** (**0.32, 0.64**)	**0.57** (**0.17, 0.98**)	**0.62** (**0.32, 0.92**)	0.23 (−0.06, 0.53)
lag1	**0.45** (**0.29, 0.61**)	0.34 (−0.06, 0.75)	**0.64** (**0.34, 0.95**)	**0.32** (**0.03, 0.61**)
lag0	**0.34** (**0.18, 0.50**)	0.10 (−0.29, 0.49)	**0.42** (**0.13, 0.72**)	**0.45** (**0.16, 0.73**)

Abbreviations: The statistically significant estimates are highlighted in bold. CHD, coronary heart disease. CVD, cardiovascular disease. lag0–3, the moving average concentration on the present day and the previous 3 days. lag0–2, the moving average concentration on the present day and the previous 2 days. lag0–1, the moving average concentration on the present day and the previous day.

**Table 3 ijerph-20-00746-t003:** ER% and 95% CIs of hospitalization risk for cardiovascular diseases associated with every 10 μg/m^3^ increase in short-term PM_2.5_ at lag0–2: stratified by single environmental factors.

Residential Exposure Factors	^a^ Level of Residential Exposure	Total CVD	Hypertension	CHD	Stroke
Greenness	Low	**0.99** (**0.71, 1.28**)	**0.87** (**0.16, 1.59**)	**1.36** (**0.84, 1.89**)	**0.84** (**0.31, 1.37**)
	High	**0.45** (**0.13, 0.77**)	0.14 (−0.58, 0.87)	**0.63** (**0.09, 1.17**)	**0.55** (**0.04, 1.06**)
Long-term PM_2.5_	Low	**0.73** (**0.41, 1.06**)	0.69 (−0.14, 1.53)	**0.67** (**0.05, 1.29**)	**0.56** (**0.01, 1.12**)
	High	**0.70** (**0.43, 0.97**)	0.32 (−0.33, 0.97)	**1.11** (**0.61, 1.61**)	**0.66** (**0.16, 1.16**)

Abbreviations: The statistically significant estimates are highlighted in bold. CHD, coronary heart disease. CVD, cardiovascular disease. lag0–2, the moving average concentration on the present day and the previous 2 days. ^a^ Low and high levels of residential exposure were divided by the median of all participants’ NDVIs within 250 m and long-term PM_2.5_ concentration (1 year).

**Table 4 ijerph-20-00746-t004:** ERs% and 95% CIs of hospitalization risks for cardiovascular diseases per 10 μg/m^3^ increase in the lag0–2 of short-term PM_2.5_, stratified by dual environmental factors.

Admissions	^a^ Long-Term PM_2.5_ Level	^b^ Residential Greenness			
		Quartile 1 (Least Green)	Quartile 2	Quartile 3	Quartile 4 (Greenest)
Total CVD	Low	**1.61** (**0.95, 2.27**)	**0.87** (**0.23, 1.52**)	0.35 (−0.30, 1.00)	0.39 (−0.30, 1.09)
	High	**0.58** (**0.04, 1.11**)	**1.19** (**0.66, 1.73**)	**0.66** (**0.13, 1.20**)	0.01 (−0.56, 0.59)
Hypertension	Low	**1.98** (**0.39, 3.60**)	−0.03 (−1.62, 1.58)	0.47 (−1.09, 2.05)	0.26 (−1.56, 2.12)
	High	0.27 (−1.01, 1.56)	1.10 (−0.19, 2.40)	0.44 (−0.84, 1.74)	−0.70 (−2.17, 0.80)
CHD	Low	**1.23** (**0.03, 2.44**)	0.79 (−0.40, 2.00)	0.66 (−0.52, 1.85)	0.74 (−0.62, 2.12)
	High	**1.17** (**0.20, 2.15**)	**1.84** (**0.87, 2.82**)	**1.04** (**0.08, 2.02**)	−0.44 (−1.55, 0.68)
Stroke	Low	**1.32** (**0.15, 2.50**)	**0.94** (**0.10, 1.78**)	0.77 (−0.39, 1.94)	0.51 (−0.65, 1.69)
	High	0.38 (−0.58, 1.34)	0.69 (−0.27, 1.66)	−0.79 (−1.95, 0.38)	0.80 (−0.17, 1.77)

Abbreviations: The statistically significant estimates are highlighted in bold. CHD, coronary heart disease. CVD, cardiovascular disease. lag0–2, the moving average concentration on the present day and the previous 2 days. ^a^ Low and high long-term PM_2.5_ values were divided by the median of all participants’ long-term PM_2.5_ concentrations (1 year). ^b^ Divided into four parts based on the quartile of all participants’ NDVIs within 250 m.

## Data Availability

The data that support the findings of this study are available from the Wuhan Information Center of Health and Family Planning, but restrictions apply to the availability of these data, which were used under license for the current study, and so are not publicly available. Data are however available from the authors upon reasonable request and with permission of the Wuhan Information Center of Health and Family Planning.

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
