# Peer review of "PM2.5-Associated Hospitalization Risk of Cardiovascular Diseases in Wuhan: Cases Alleviated by Residential Greenness"

_ijerph, 2022, doi:10.3390/ijerph20010746_

Round 1

Reviewer 2 Report

Please find the comments in the attached file. 

Round 2

Reviewer 2 Report

The effort of the authors are duly appreciated, and the manuscript has been improved substantially. Moreover, the authors have addressed the comments adequately. However, further language editing is required as many linguistic inconsistencies are persistent.

Specific comment

L97: Wuhan, a megacity in central China, had a land area of 8,569.15 km2 and a population 97 of more than 10 million Replace “had with “with

L118: previous study.” Which study/studies?

(3) Many abbreviations, such as CCO, DF, df, etc., remain undescribed

Author Response

Thank you for your valuable comments on the language and abbreviation, which will help improve the clarity and readability of our manuscript. In the latest version, we have revised the corresponding contents specified in your comments. Besides, we have carefully checked and revised some language and grammar problems to improve the readability. (Line 35, 36, 50, 73, 76, 84, 85, 96, 101, 115, 119, 124, 146, 179, 187, 192, 196, 205, 206, 260, 269, 283, 285, 289, 303, 309, 313, 323,325,326, 341, 344, 379, 399).